# Advances in Cholinesterase Inhibitor Research—An Overview of Preclinical Studies of Selected Organoruthenium(II) Complexes

**DOI:** 10.3390/ijms25169049

**Published:** 2024-08-21

**Authors:** Monika C. Žužek

**Affiliations:** Institute of Preclinical Sciences, Veterinary Faculty, University of Ljubljana, Gerbičeva 60, 1000 Ljubljana, Slovenia; monika.zuzek@vf.uni-lj.si

**Keywords:** cholinesterases, enzyme inhibition, organoruthenium complex, neuromuscular junction, mouse

## Abstract

Cholinesterase (ChE) inhibitors are crucial therapeutic agents for the symptomatic treatment of certain chronic neurodegenerative diseases linked to functional disorders of the cholinergic system. Significant research efforts have been made to develop novel derivatives of classical ChE inhibitors and ChE inhibitors with novel scaffolds. Over the past decade, ruthenium complexes have emerged as promising novel therapeutic alternatives for the treatment of neurodegenerative diseases. Our research group has investigated a number of newly synthesized organoruthenium(II) complexes for their inhibitory activity against acetylcholinesterase (AChE) and butyrylcholinesterase (BChE). Three complexes (C1a, C1-C, and C1) inhibit ChE in a pharmacologically relevant range. C1a reversibly inhibits AChE and BChE without undesirable peripheral effects, making it a promising candidate for the treatment of Alzheimer’s disease. C1-Cl complex reversibly and competitively inhibits ChEs, particularly AChE. It inhibits nerve-evoked skeletal muscle twitch and tetanic contraction in a concentration-dependent manner with no effect on directly elicited twitch and tetanic contraction and is promising for further preclinical studies as a competitive neuromuscular blocking agent. C1 is a selective, competitive, and reversible inhibitor of BChE that inhibits horse serum BChE (hsBChE) without significant effect on the peripheral neuromuscular system and is a highly species-specific inhibitor of hsBChE that could serve as a species-specific drug target. This research contributes to the expanding knowledge of ChE inhibitors based on ruthenium complexes and highlights their potential as promising therapeutic candidates for chronic neurodegenerative diseases.

## 1. Introduction

Cholinesterase (ChE) inhibitors remain important therapeutic agents for the symptomatic treatment of some chronic neurodegenerative diseases associated with functional disorders of the cholinergic system. They are also used to treat glaucoma and prevent optic neuropathy, as well as for the symptomatic treatment of the autoimmune disease myasthenia gravis (MG).

During the past two decades, significant research efforts have been made to broaden our understanding of the complex molecular pathogenesis of Alzheimer’s disease (AD) and to provide a robust framework for the development of effective pharmacological treatment approaches. Multiple hypotheses have been proposed to explain the multifaceted nature of the disease, the most prominent ones being the amyloid hypothesis, the cholinergic hypothesis, the tau propagation hypothesis, the calcium homeostasis hypothesis, the neurovascular hypothesis, the inflammatory hypothesis, the metal ion hypothesis, the lymphatic system hypothesis, and the mitochondrial dysfunction hypothesis [1]. The cholinergic hypothesis revolutionized the field of AD and led to the development of ChE inhibitors as one of the few clinically proven drug therapies for the treatment of AD. In the early stages of the disease, the expression of the acetylcholinesterase (AChE) is elevated in some areas of the brain. However, as the disease progresses, the level of AChE in the brain begins to decrease (up to an 85% less than normal), while the expression of the butyrylcholinesterase (BChE) increases (up to a 120% more than normal). BChE plays a role in the later stages of AD, as it also contributes to the increased breakdown of acetylcholine (ACh) [2,3].

Although ChE inhibitors do not cure AD, they improve cognitive functions and diminish the clinical signs of AD. Thus, they remain the standard pharmacological approach for treating both the cognitive and functional symptoms of AD at various stages of the disease [4]. While these drugs improve the quality of life of AD patients, they have no significant effect on the occurrence or progression of AD and have several adverse effects [1]. Only a handful of ChE inhibitors are currently approved for the symptomatic treatment of AD, and significant research efforts are being directed toward the development of novel ChE derivatives and ChE inhibitors with novel scaffolds to enhance their efficacy and safety profile. Current ChE inhibitors primarily target AChE; some also inhibit BChE, albeit with a much lower inhibitory effect [5]. However, considering the complex and dynamic roles of both ChE in AD, it is prudent to focus on developing compounds that simultaneously inhibit both enzymes. This dual inhibition approach could more effectively address the evolving cholinergic deficits throughout the progression of AD [2].

Advances in drug design and high-throughput compound libraries have facilitated the development of novel derivatives and ChE inhibitors with novel scaffolds as well as the development of multitarget-directed ligands [6,7,8,9,10].

In the last decade, ruthenium compounds have emerged as promising therapeutic alternatives to conventional drugs for treating neurodegenerative diseases [11]. Various ruthenium complexes exhibit a wide range of biological activities, including antitumor, antimicrobial, and immunosuppressive effects [12,13,14,15,16]. Some ruthenium complexes, including organoruthenium(II) arene complexes, also act as ChE inhibitors [17,18] and/or as inhibitors of amyloid-β (Aβ) peptide aggregation [18], thus combining their potential neuroprotective and anticholinergic effects [11].

This review aims to summarize the literature pertaining to ChE inhibitors with a focus on preclinical studies of select ruthenium compounds that also function as ChE inhibitors.

## 2. Cholinergic Nervous System

The cholinergic nervous system encompasses the parts of the nervous system that utilize ACh as a neurotransmitter. This system includes cholinergic neurons, which are present in both the central and peripheral nervous systems [19,20]. Naturally occurring ACh was first discovered by H. Dale [21]. This neurotransmitter plays a role in memory, learning, attention, arousal, and involuntary smooth and voluntary skeletal muscle movement. It is synthesized in cholinergic neurons by the acetylation of choline in the presence of the enzymes choline acetyltransferase and acetyl coenzyme A [22,23,24]. Neurons are not the only cells that synthesize ACh. Its synthesis can also occur in other cell types, including those in the skin, kidneys, eyes, liver, placenta, T lymphocytes, and supporting cells of the central nervous system, particularly astrocytes, as they all express the enzyme choline acetyltransferase [24,25]. In the brain, ACh originates from two major regions: the basal forebrain and the mesopontine tegmentum. ACh is stored in synaptic vesicles in cholinergic neurons. Two ACh reserves can be distinguished: the smaller reserves are located near the active zones, which are membrane-bound and can be released immediately [26], whereas the larger reserves are formed by ACh molecules that cannot be released directly and are either free in the cytoplasm or stored in free vesicles. Each synaptic vesicle contains and releases up to 10,000 ACh molecules, which is also referred to as a quantum [26,27]. Depolarization of the membrane of cholinergic neurons is followed by the opening of voltage-gated calcium channels in the nerve terminal, followed by an influx of Ca^2+^ into the nerve terminal. These increased Ca^2+^ levels trigger the exocytosis of ACh vesicles, with the vesicles fusing with the presynaptic membrane in the active zones. Four proteins are involved in the fusion of vesicles and exocytosis: synaptotagmin, synaptobrevin, syntaxin, and SNAP-25 [28,29]. The ACh molecules thus released into the synaptic cleft diffuse to the postsynaptic membrane, where they bind to (i) ionotropic nicotinic acetylcholine receptors (nAChRs) or (ii) metabotropic G protein-coupled muscarinic acetylcholine receptors (mAChRs) [30]. The agonistic effect of a single ACh molecule that binds to the receptor is short-lived, lasting only a few milliseconds, since the enzyme acetylcholinesterase (EC 3.1.1.7) promptly hydrolyzes the ACh molecule to acetate and choline. Each AChE molecule can break down up to 25,000 ACh molecules per second [31,32,33].

## 3. Cholinesterases

AChE and BChE, members of the serine hydrolase superfamily of enzymes, hydrolyze the ACh; however, they have different substrate specificities and tissue distributions [34,35]. AChE is located in the synapses of the central and peripheral nervous system in both motor and sensory nerve fibers. In addition, it is also found in the motor end plate, in the cholinergic synapses of the autonomic nervous system (preganglionic sympathetic and parasympathetic fibers and postganglionic parasympathetic fibers), and in a bound form on erythrocyte membranes [36,37]. BChE is a nonspecific ChE that degrades choline and noncholine esters. It is present in the blood plasma, liver, pancreas, and at the neuromuscular junction and hydrolyzes the artificial substrate butyrylcholine faster than ACh [36,38]. Its endogenous substrate has not yet been identified, and its actual physiological role has remained unclear for many decades [39,40]. BChE is thought to have evolved from AChE to function as a general detoxifier, acting as a protective bioscavenger of bioactive esters. Additionally, BChE is thought to provide redundancy of ChE activity in the neurotransmission process [38,41]. In the brain, BChE is expressed especially in glial cells, but it can also be found in endothelial cells and neurons, where, in contrast to AChE, it is mostly present outside the synaptic cleft [42]. BChE expression exceeds that of AChE in many tissues and is present at much lower levels in the brain, skeletal muscle, and peripheral nerves [43].

## 4. Cholinesterase Inhibitors

ChE inhibitors are a diverse group of compounds with different structures, activities, and origins that inhibit the activity of ChEs [36,44]. The result is a decreased rate of ACh degradation and an increased amount of synaptic ACh available for nAChR and mAChR stimulation in the central and peripheral nervous system [45]. ChE inhibitors can be divided on the basis of their specificity to nonspecific ChE inhibitors (affecting both AChE and BChE) and specific inhibitors (AChE- or BChE-specific). AChE has three binding sites for inhibitory ligands: the acyl pocket, the choline pocket, and the peripheral anion site (PAS) [46,47]. These distinct domains contribute to the differences in specificity between AChE and BChE. AChE has a more pronounced PAS and a narrower gorge than BChE. In general, aromatic compounds have a greater affinity for AChE than BChE, as the aromatic amino acid residues are replaced by aliphatic ones in the BChE gorge [48].

ChE inhibitors are divided into three classes based on their mode of action: reversible, irreversible, and pseudoreversible [49]. Reversible inhibitors are further subdivided into competitive and noncompetitive types [50]. Reversible and some pseudoirreversible ChE inhibitors are most commonly used for therapeutic and/or diagnostic purposes, e.g., in AD, MG, Parkinson’s disease (PD), Lewy body dementia, vascular dementia, traumatic brain injury, bladder distension, postoperative ileus, and glaucoma [19,36,51,52,53]. On the other hand, irreversible ChE inhibitors are commonly used as pesticides or chemical warfare agents. Selective irreversible and pseudoreversible AChE inhibitors can also have therapeutic applications in some conditions, and some organophosphorus compounds are used for the local treatment of chronic glaucoma [36].

## 5. Clinical Application of ChE Inhibitors in AD

ChE inhibitors, which prolong the lifetime of ACh in synapses, improve cognitive function, and mitigate the clinical symptoms of AD, thus remain standard, cornerstone pharmacological approaches in the symptomatic treatment of AD at various stages of the disease [4]. As studies have shown that the expression and function of both AChE and BChE are altered in AD [3], it is essential to develop new compounds that can either inhibit both enzymes simultaneously or selectively target BChE [1].

The clinical usefulness of ChE inhibitors is limited by various adverse effects, which are dose-dependent. The adverse effects of ChE inhibitors are dependent on their mechanism of action and can include the following: (i) activation of mAChR in the central nervous system (e.g., tremor, hypotension, bradycardia); (ii) mixed central and peripheral activation of mAChR (e.g., sweating, nausea, emesis, bradycardia); (iii) peripheral activation of mAChR (e.g., diarrhea); and (iv) overstimulation/blockade of nAChR in neuromuscular junctions (e.g., muscular weakness) [54,55]. Substances that inhibit AChE can also block nAChRs in the central and peripheral neuromuscular systems and may cause paralysis of skeletal muscle fibers [56,57].

To date, the European Medicines Agency (EMA) has approved only four centrally acting ChE inhibitors for the symptomatic treatment of AD: tacrine (1995), donepezil (1998), rivastigmine (1998), and galantamine (2000) [58]. Tacrine, a potent, noncompetitive, nonselective, reversible AChE inhibitor, was withdrawn from the market in 2012 because of its poor bioavailability and side effects [59]. Currently, only donepezil, galantamine, and rivastigmine are ChE inhibitors used for clinical purposes, particularly for the symptomatic treatment of mild to moderate AD [60] (Figure 1).

## 6. Advances in the Development of Novel ChE Inhibitors for the Treatment of AD

ChE inhibitors are a diverse group of chemical compounds with various structures, activities, and origins. ChE inhibitors can be classified based on their origin into natural, hybrid, and synthetic compounds. ChE inhibitor drugs currently in clinical use for the treatment of AD are synthetically derived; however, they mostly originate from natural compounds, mainly alkaloids (e.g., galantamine and rivastigmine) [9]. In recent years, advances in computer-aided drug design and high-throughput compound libraries have facilitated the development of novel derivatives and ChE inhibitors with novel scaffolds [8]. Recently, a number of reviews summarizing extensive research into novel ChE inhibitors as well as the development of multitarget-directed ligands, including ChE inhibitors, have been published. These inhibitors, derived from natural, synthetic, or hybrid sources, have been developed using both conventional methods and computer-aided drug design and are mostly at a preclinical stage [6,7,9,10].

One promising approach for the development of novel drugs for AD focuses on creating organometallic complexes. Many metal ions, particularly transition metals such as ruthenium, exhibit multiple oxidation states under physiological conditions that allow them to participate in redox reactions. Modifying substituents or coordinated ligands can alter the properties of organometallic complexes, resulting in significant pharmacological changes such as enhanced target specificity and selectivity, changes in lipophilicity, etc. Furthermore, the properties of organometallic complexes can be fine-tuned by attaching a selective and tailored organic ligand of known therapeutic value [61,62].

Organometallic complexes have emerged as potential promising therapeutic agents for AD as symptomatic treatment through ChE inhibition as well as disease-modifying treatment by targeting key pathological processes through amyloid-beta (Aβ) directed complexes [63,64]. In the last five years, several studies have investigated organometallic complexes containing metals such as platinum [65,66], ruthenium [67,68,69], copper [70,71,72,73,74,75], iron [69,72], zinc [72,73,76,77], cobalt [72,74], and palladium [78] as ChE inhibitors.

Several organoruthenium complexes have been investigated for their potential use as ChE inhibitors. A number of research groups investigated various ruthenium polypyridyl complexes as AChE inhibitors with IC_50_ in the submicromolar or low micromolar range. Mulcahy et al. have found a potent complex (IC_50_ of 200 nM) in their study of ruthenium polypyridyl complex library [79]. In the last decade, Vyas et al. investigated a number of ruthenium polypyridyl complexes that act as inhibitors of Aβ aggregation as well as AChE inhibitors in the submicromolar range [18,80] or are weak inhibitors of AChE [81]. In their study of a library of ruthenium(II) polypyridyl complexes, Alatrash et al. found a number of AChE inhibitors with IC_50_ in the low micromolar range (<15 μM) [82]. Recently, Almeida et al. have also investigated a ruthenium(II) polypyridyl complex with an estimated IC_50_ value of 39 μM for AChE [67]. Boubakri et al. reported a complex with AChE inhibitory activity in their study of a series of ruthenium(II) complexes with N-heterocyclic carbene ligands [83].

Fewer research groups have reported on BChE or combined AChE/BChe inhibitory activity of organoruthenium complexes with IC_50_ in the submicromolar or low micromolar range. Silva et al. described two ruthenium complexes that inhibit both ChE, with greater activity against AChE (IC_50_ 9.64 and 8.72 μM) than against BChE (IC_50_ 50.34 and 39.75 μM) [84]. Kladnik et al. have reported on organoruthenium(II) chlorido complexes with methyl-substituted pyrithione analogues or bicyclic aromatic pyrithione analogues that inhibit both ChE in the low micromolar range, with IC_50_ values for AChE from 4.9 μM to 14.3 μM and 0.2 to 2.7 μM for BChE [68]. Kosinska et al. have reported on ruthenium aminophosphonate complexes with either moderate inhibitory activity against AChE (lowest IC_50_ 164 μM) or BChE (lowest IC_50_ 186 μM) [69].

In addition to ChE inhibition, organoruthenium complexes have been designed to target Aβ peptides, inhibit tau aggregation, act as chelators, and may serve as multitarget agents [18,63,80,81,85,86,87,88,89,90].

## 7. Preclinical Study of the Effects of Natural and Synthetic Cholinesterase Inhibitors as Potential Therapeutic Agents: Our Research Contribution to the Field of Preclinical ChE Inhibitors

For almost two decades, our laboratory has been studying the effects of various natural and synthetic ChE inhibitors on mammalian cholinergic systems at the preclinical level in vitro, ex vivo, and in vivo. Among natural compounds, protoberberine alkaloids from *Papaver setiferum*, discorhabdine alkaloids (B, G, L), dehydrodiscorhabdin C from Antarctic *Latrunculia* spp., and polymeric alkylpyridinium salts isolated from crude extracts of the Mediterranean marine sponge *Reniera sarai* have been studied [91,92,93]. Among synthetic analogs, we have studied several compounds on the basis of the structures of ChE-inhibitory alkaloids from marine sponges [19,56,94,95,96,97], whereas in recent years, our work has focused mostly on evaluating the activity of organoruthenium(II) complexes.

Our work on ChE inhibitors is based on the identification and characterization of their ChE inhibitory activity and the type of inhibition. Since ChE inhibitors may cause undesirable or side effects in the cholinergic system, we have been investigating the effects of the selected ChE inhibitors on peripheral neuromuscular transmission in isolated skeletal muscles, mostly on mouse hemidiaphragms (muscle twitch and recordings of resting membrane potentials (rVm), end-plate potentials (EPPs), and miniature end-plate potentials (MEPPs)) or in vivo effects in rats (monitoring of arterial blood pressure, respiratory activity and electrocardiograms, and histopathological evaluation). These effects in vivo may be due to activation of mAChRs in the central nervous system, mixed central and peripheral mAChR activation, peripheral mAChR activation, and/or overstimulation/inhibition of nAChRs in the neuromuscular junction [54]. Some ChE inhibitors have dual inhibitory effects, where they inhibit ChEs and nAChRs in the neuromuscular system and cause paralysis of skeletal muscle fibers [56,57]. With this in mind, we are investigating the effects of this type of compound on different types of nAChRs (neuronal- and muscle-type nAChRs) using a two-electrode voltage clamp on *Xenopus laevis* oocytes with muscle-type nAChRs microimplanted into their membrane or expressed as neuronal-type nAChRs before in vivo experiments. Over the past five years, much of our work in the field of ChE inhibitors has focused on finding potent ChE inhibitors from the library of organoruthenium(II) complexes, the effects of which we will focus on in the following sections. A library of different organoruthenium(II) complexes has been synthesized by the Turel research group (University of Ljubljana, Slovenia) (http://ruturel.fkkt.uni-lj.si/).

### 7.1. Structures of the Tested Organoruthenium(II) Complexes

Different ruthenium complexes have different biological effects, including anti-ChE activity, which are strongly influenced by the oxidation state of the ruthenium ion (usually +2, +3, and +4), the number and type of ligands, and the coordination geometry. In recent years, we tested two groups of organoruthenium(II) complexes: (i) organoruthenium(II) complexes with *p*-cymene as an arene ligand and (ii) one organoruthenium(II) carbonyl complex as a CO-releasing molecule (CORM). Arene ligands in the structure of organoruthenium(II) arene complexes affect the electron distribution of the complex and increase its hydrophobicity. The electron distribution affects the stability of the complex, whereas increased hydrophobicity facilitates passage through lipid membranes. The physicochemical properties of ruthenium(II) arene complexes and, consequently, their biological activities are particularly influenced by the ligands at the X, Y, and Z positions (Figure 2) [17,98]. Organoruthenium(II) arene complexes have been developed that carry both chelating (pyrithione, β-diketonate, methoxypyridine, and nitrophenantroline) and monodentate ligands (Cl^−^, Br^−^, I^−^, and 1,3,5-triaza-7-phosphaadamantane—pta). The three-dimensional structure and good fit of the complexes in the active sites of ChEs improve coordination with amino acid residues and determine their selective action [17,98].

### 7.2. Ruthenium Complexes as Cholinesterase Inhibitors

The three organoruthenium(II) complexes with O,S-ligand pyrithione with various monodentate halide ligands (Cl^−^, Br^−^, and I^−^) were evaluated for their ChE inhibitory effects on electric eel (ee) AChE, human recombinant (hr) AChE, and horse serum (hs) BChE. The inhibitory activity of all three complexes was in the micromolar range (IC_50_ values between 1.9 and 13.14 µM) for all tested ChE. However, some important differences were noted, such as the greater susceptibility of hrAChE to Br^−^ and I^−^ analogs than to Cl^−^ and the greater susceptibility of BChE to AChE. The higher susceptibility of hrAChE to Br^−^/I^−^ analogs than to Cl^−^ analogs (**C1a**) (Figure 3A) was due to the lower sensitivity of hrAChE used in the previous study, where structural analysis revealed that half of the molecules in the crystal were blocked by the peptide loop formed by amino acid residues 483–491 [47]. Repeated tests assessing the inhibitory potential of the Cl^−^ analog toward the hrAChE lot used in the study confirmed that the observed differences were due to structural defects in the enzyme rather than the presence of various monodentate Z ligands (Cl^−^, Br^−^, and I^−^), and all three analogs had comparable inhibitory activities for the ChEs used in the study. The inhibition of all three ChEs was reversible and competitive, and the K*i* values were in the micromolar range. Based on the type of inhibition observed, it is probable that these inhibitors interact with the active site within the enzyme gorge. Another organoruthenium(II) pyrithione complex with the pta ligand (Figure 3A) effectively inhibited only BChE, with an IC_50_ value of 0.5 µM. Its IC_50_ value is about 15- and 7-fold lower than that of pyrithione complexes with monodentate halide ligands. This complex shows that changes in Z-ligands can lead to changes in inhibitory activities or changes in specificities toward the enzymes used [17,100].

Another group of three complexes studied were organoruthenium(II) complexes with β-diketonates with different substituents and the monodentate Z ligands Cl^−^ or pta (Figure 3C). Inhibition assays showed that these complexes effectively inhibited only BChE in the micromolar range, with IC_50_ values between 19.2 and 31.99 µM, respectively. BChE inhibition by these complexes was apparently of a reversible competitive type, indicating that the complexes interact with the active site within the ChEs. The results showed that different substituents of β-diketonates in the organoruthenium(II) chlorido complexes had no effect on inhibitory activities or on the specificities against the ChEs. However, replacement of the Z ligand Cl^−^ with pta resulted in a slight increase in the inhibitory activity against BChE, suggesting that the nature of the Z ligand in the organoruthenium(II) β-diketonate complexes has a minor effect on the inhibitory activity of the complex [17].

Two organoruthenium(II) complexes with methoxypyridine as chelating ligand and Cl^−^ or pta (**C1**) (Figure 3D) as monodentate ligand were also synthesized. The chlorido complex unselectively inhibited all tested enzymes in the micromolar range (IC_50_ between 1.71 and 6.56 µM). In contrast to the chlorido complex, the pta complex selectively inhibited BChE with an inhibition potential within a pharmaceutically relevant range (IC_50_ value of 2.88 µM). These inhibitions were all of the reversible competitive type. Comparing the pta complex with its chlorido analog, we found that the selectivity of the complex toward BChE is due to the addition of the pta ligand [101].

We also investigated the inhibitory potential of the organoruthenium(II) complex with nitrophenantroline as a chelating ligand and Cl^−^ as a monodentate ligand (**C1-Cl**) (Figure 3B) toward ChE enzymes. The complex inhibits ChE enzymes nonselectively, with IC_50_ values ranging from 16.1 to 26.2 μM. Our results also show that the nitrophene ligand is a stronger inhibitor of eeAChE and hsBChE than the organoruthenium complex with nitrophenantroline, though it is less effective against hrAChE. The combined results showed that nitrophene had even better anti-ChE activity in some cases, but unlike the complex, the ligand is not soluble under physiological conditions [102].

In the library of complexes, we have also included a CORM (Figure 3E) complex with the pyrithione ligand. This decision stems from the literature indicating the protective role of low concentrations of CO in the central nervous system, suggesting a beneficial effect in diseases such as AD, traumatic brain injury, and stroke [103]. If the complex simultaneously inhibits ChEs and releases CO, it could have a dual beneficial effect in the treatment of AD. However, this complex had no effect on the activity of ChEs [17].

### 7.3. Effects of Selected Organoruthenium(II) Complexes on Skeletal Muscle Function and Neuromuscular Transmission

#### 7.3.1. Organoruthenium(II) Pyrithione Chlorido Complex (C1a)

This nonselective ChE inhibitor was tested at high concentrations (38, 113, and 227 µM) to investigate its adverse effects on the neuromuscular system. At a concentration of 38 µM, **C1a** had no effect on the maximal amplitudes of single and tetanic twitches, whereas a time-dependent partial block of single twitches and tetanic contractions was observed at the higher **C1a** concentrations. Inhibition of AChE in the neuromuscular junction is associated with the amplitude drop of tetanic contractions produced by repetitive high-frequency stimulation of the motor nerve. Tetanic fade was observed only at the higher **C1a** concentrations, as discussed above, suggesting AChE inhibition in the motor end plate. We also examined the effects of a complex on membrane potentials. **C1a** at a concentration of 38 µM had minimal effects on evoked neurotransmitter release, as shown by the slightly decreased EPP amplitude after 60 min of exposure. After the first 30 min of exposure, we observed a prolongation of the EPP decay half-life time suggesting AChE inhibition, but with no effect on tetanic muscle contraction. At the higher concentrations (113 and 227 µM), the complex increased rVm, decreased EPP amplitudes, and decreased the amplitude and half-life decay times of MEPPs. These effects could be responsible for the decrease in the amplitude of muscle contraction at these concentrations [100].

#### 7.3.2. Organoruthenium(II) Nitrophenantroline Chlorido Complex (C1-Cl)

The effects of the ChE inhibitory complex **C1-Cl** on neuromuscular transmission were determined in a concentration range from 5 to 40 μM. We found that **C1-Cl** progressively and reversibly decreased the amplitude of nerve-evoked isometric muscle twitch and nerve-evoked tetanic contraction in a concentration-dependent manner without any noticeable change in their decline phase. At a concentration of 20 μM, the complex reduced the amplitude of the nerve-evoked muscle twitch by approximately 50%. The IC_50_ values for the nerve-evoked single twitch and tetanic contraction were 19.44 μM and 9.68 μM, respectively. In contrast to the nerve-evoked, directly elicited single twitch and tetanic contraction were not affected at the same or even at the highest (40 μM) **C1-Cl** concentration. At a concentration of 3 μM, the well-known AChE inhibitor neostigmine caused a rapid and almost complete recovery of single twitch block produced by the **C1-Cl** complex. This result strongly suggests that **C1-Cl** does not inhibit AChE but causes the competitive block in the neuromuscular junction. The complex at given concentrations did not affect the rVm but, in a concentration-dependent manner, decreased EPP and MEPP amplitude without a significant decrease in MEPP frequency at a low (5 μM) concentration. The concentration-dependent block of EPP amplitude below the critical level for action potential generation may explain the observed concentration-dependent nerve-evoked block of muscle twitch and tetanic muscle contraction produced by **C1-Cl**. Nevertheless, in measurements of the decay phase of synaptic potentials, we also failed to observe any prolongation of the decay phase of EPPs and MEPPs, a parameter relevant to anti-ChE activity. Taken together, the results indicating a postsynaptic effect on muscle-type nAChR receptors include the reduction of the amplitude of spontaneous MEPPs, a stable rVm of muscle fibers, and the blocking of EPPs without altering the passive membrane properties of muscle fibers [102].

#### 7.3.3. Organoruthenium(II) Methoxypyridine Pta Complex (C1)

This selective BChE inhibitory complex was tested at concentrations of 30, 60, 90, and 120 µM to investigate its adverse effects on the neuromuscular transmission. We were able to demonstrate that none of the tested C1 concentrations significantly affected the amplitude of muscle contraction. To rule out any negative effects of the highest C1 concentration (120 µM) on neuromuscular transmission, we also performed measurements of rVm, EPPs, and MEPPs. During the 60 min exposure to this concentration, no effect on rVm was detected, the amplitudes of MEPPs were significantly reduced without affecting their frequency, and notably no effect on the amplitudes of EPPs was observed. Additionally, it shortened the half-decay time of both MEPPs and EPPs [101]. The reduced MEPP amplitude and the shortened half-decay time of both MEPPs and EPPs indicate a potential mild inhibitory effect of the complex on muscle-type nAChRs [104]. None of the changes characteristic for BChE inhibition in the neuromuscular junction, such as reduced frequency of MEPPs and giant MEPPs, were observed [38]. Our findings suggest that **C1** does not inhibit BChE in the neuromuscular junction or that any potential effect is overshadowed by an effect of **C1** on nAChRs. The integrated results of the measurements of muscle contraction and membrane potential parameters suggest that the supra-pharmacological concentration of **C1** (120 µM) does not significantly influence the neuromuscular transmission and skeletal muscle physiology [101].

## 8. Conclusions

Cholinesterase inhibitors have been used for decades for the symptomatic treatment of several chronic neurodegenerative diseases associated with functional disorders of the cholinergic system, particularly AD. Significant research efforts have been made to develop novel derivatives and ChE inhibitors with novel scaffolds as well as multitarget-directed ligands to enhance their efficacy and safety. Organometallic complexes have emerged as promising and versatile therapeutic agents with potential applications as symptomatic treatment through ChE inhibition as well as disease-modifying agents. Among these, ruthenium complexes have gained attention in the last decade as promising novel therapeutic alternatives to conventional drugs in the treatment of neurodegenerative diseases.

Our group has contributed to the development of this field over the last ten years with our preclinical studies on various anticholinesterase substances of natural and artificial origin. Preclinical research, such as electrophysiological studies on ChE inhibitors, helps elucidate their impact on the peripheral cholinergic system. These studies provide additional insights into the therapeutic potential of ChE inhibitors and their side effects, enhancing our understanding of how these compounds can be used effectively in treating neurodegenerative diseases while identifying possible adverse effects.

Over the past decade, ruthenium complexes have emerged as promising novel therapeutic alternatives to conventional drugs in the treatment of neurodegenerative diseases. In the last five years, our group has collaborated on preclinical research on ruthenium complexes and has investigated a number of compounds from the library of newly synthesized organoruthenium(II) complexes with p-cymene as the arene ligand, along with one organoruthenium(II) carbonyl complex, CORM, for their inhibitory activity against AChEs and BChEs of human and animal origin. We have established that the anticholinesterase activity of these organoruthenium(II) complexes is strongly influenced by the nature of the ligand and that these complexes are promising candidates for further preclinical testing as ChE inhibitor drugs. The most promising potential anticholinesterase drugs or nAChR inhibitors are complexes C1a, C1, and C1-Cl. The organoruthenium C1a can reversibly inhibit the activities of both AChE and BChE within a pharmaceutically relevant range. As it has no undesirable peripheral effects, C1a can be proposed as a promising candidate for the treatment of AD. The organoruthenium complex C1-Cl reversibly and competitively inhibits ChEs, especially AChE, in a pharmaceutically relevant range. While the anti-AChE effect of C1-Cl at the neuromuscular junction is not evident, C1-Cl inhibits nerve-induced skeletal muscle twitch and tetanic contraction in a concentration-dependent manner with no effect on directly elicited twitch and tetanic contraction. The most likely mechanism of its inhibitory effect is the reversible inhibition of muscle-type nAChRs. Therefore, C1-Cl is also interesting for further preclinical studies as a new competitive neuromuscular blocking drug. As a selective, competitive, and reversible inhibitor of BChE, the organoruthenium(II) complex C1 inhibits horse BChE in the micromolar range but does not inhibit human BChE or canine BChE in the pharmaceutically relevant range. C1 has no significant effect on the peripheral neuromuscular system and is likely a highly species-specific inhibitor that could serve as a species-specific drug target.

## Figures and Tables

**Figure 1 ijms-25-09049-f001:**
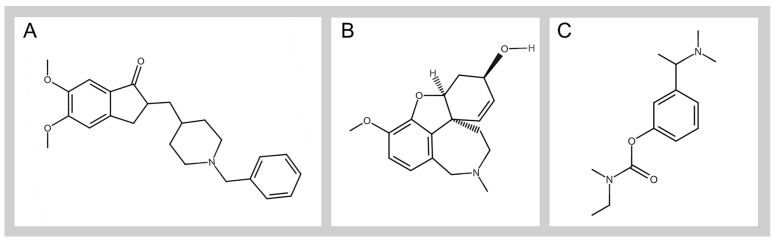
General structure of donepezil (**A**), galantamine (**B**), and rivastigmine (**C**). MarvinJS was used to draw the chemical structures, Marvin 24.3.0 (re389bc4da912), 2024, ChemAxon (http://www.chemaxon.com, accessed on 1 August 2024).

**Figure 2 ijms-25-09049-f002:**
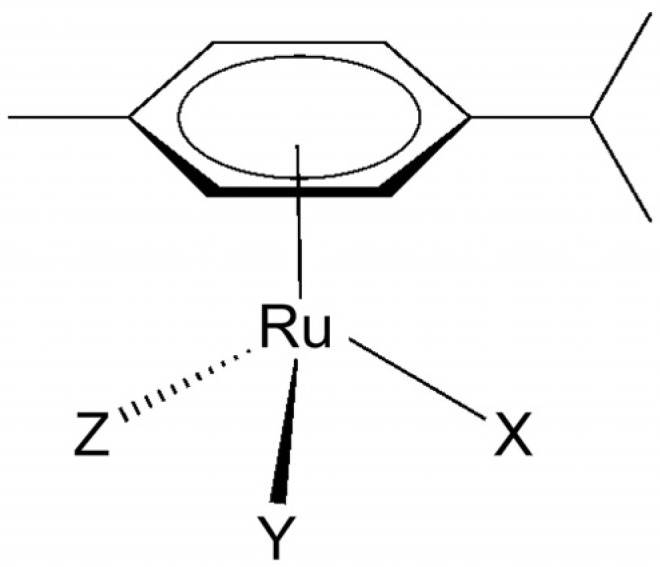
General structure of ruthenium(II) cymene complexes [99]. Different chemical groups (X, Y, and Z) allow for the preparation of diverse chemical libraries.

**Figure 3 ijms-25-09049-f003:**
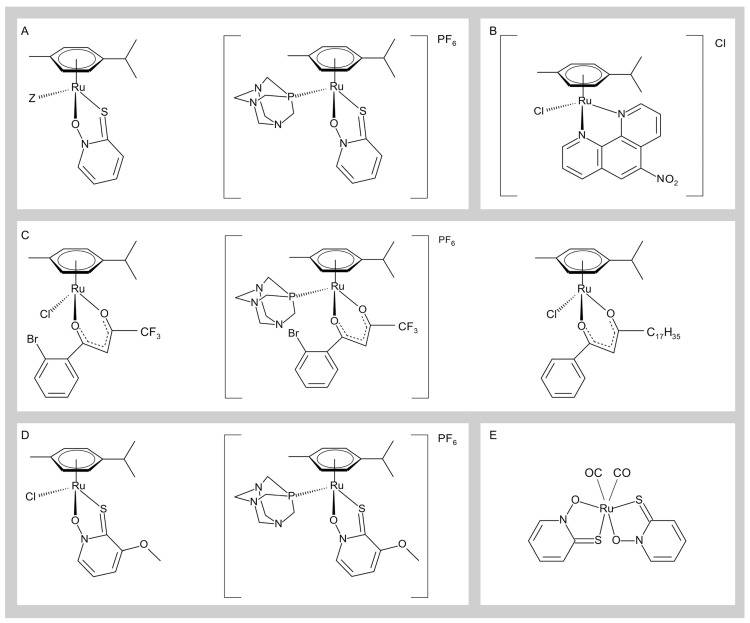
Organoruthenium(II) (**A**) pyrithione complex with halide ligand (Z = Cl^−^(**C1a**), Br^−^, I^−^) or pta; (**B**) nitrophenantroline complex with Cl^−^ (**C1-Cl**); (**C**) β-diketonate complex with Cl^−^ or pta; (**D**) methoxypyridine complex with Cl^−^ or pta (**C1**); and organoruthenium(II) carbonyl complex (**E**).

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
