# Peer review of "Advances in Cholinesterase Inhibitor Research—An Overview of Preclinical Studies of Selected Organoruthenium(II) Complexes"

_ijms, 2024, doi:10.3390/ijms25169049_

Round 1
Reviewer 1 Report
Comments and Suggestions for Authors
The present review describes preclinical advances in the development of new Cholinesterase inhibitors while revisiting and comparing to some of the FDA approved ChE. It seems that significant advances have been made in characterization of new compounds, but also that considerable work is still required to ensure that new compounds can be safely suggested for treatment of neurological and muscular disorders. However, the review is a positive incremental addition to understanding ongoing advances and should be of use to readers interested in the development of ChE inhibitors.
A few suggestions follow here:
Line 18: change “making it a promising candidate AD treatment” to “making it a promising candidate for AD treatment”
Line 59 to 62: The first 2 sentences in these lines are somewhat incongruent because cholinergic synapses and cholinergic neurons are part of BOTH the central and peripheral nervous systems, not just the autonomic nervous system (so the first sentence is not entirely correct).
The review and its abstract mention Alzheimer’s disease (AD) as a major justification for developing the targeted compounds and for the actual writing of this review. However, the title of the article and the review itself deal mainly with the development of the inhibitors in different contexts, so the starting emphasis on AD in the abstract and introduction is some what misleading. This comment does not imply that AD related therapeutic and mechanistic discussions in the review are not needed or inadequate. It is just a matter of emphasis and organization of the article.
Comments on the Quality of English LanguagePlease, see main text of review.
Reviewer 2 Report
Comments and Suggestions for Authors
This review article contains a fairly focussed and well-written account of the development of ruthenium-based cholinesterase inhibitors. These are important therapeutics due to the key role of cholinesterase in the progress of neurodegenerative illnesses, notably Alzheimer's disease.
The review is well structured and contains a very good introduction to cholinesterases which places the subject into appropriate context. The key classes of inhibitors are then described, again divided into chemical classes and with a detailed description of their methods of development and modes of action. Three of the most promising Ru-based organometallic compounds are illustrated.
The review is very well written and contains a large number of references - 128 in total, which serve as a representative sample of this important area of current global research work.
My only fairly minor criticism is that the review lacks illustrations which would very much help to communicate the results. I would recommend that Chemdraw illustrations of the various compounds which are discussed, are added. Notably these should include Donepezil, Galantamine, a representative carbamate, rivastigmine and neostigmine. In the discussion of the Ru-based complexes, again more diagrams of the classes of Ru complex (6-7 examples) which have been studied, i.e. representing a larger set of what has already been reported, should be included.
With these additions, my view is that the review will be more accessible and will communicate the subject more clearly.
Reviewer 3 Report
Comments and Suggestions for Authors
In the submitted manuscript, Dr. Žužek summarized the findings of her research group regarding the investigation of Ru complexes as cholinergic enzyme inhibitors. The author described the effects of three Ru complexes, focusing on their inhibition potency, the mechanism of inhibition, and their influence on skeletal muscle function and neuromuscular transmission.
The manuscript contained an excessive amount of well-known information (two pages) about cholinergic enzymes, their structures, and commercially available inhibitors. This information is not useful in a paper intended for publication in 2024 and should be removed. Instead, the author should add new findings (from the last five years) about the investigation of metal complexes for Alzheimer's disease (AD), with an emphasis on their potential to inhibit cholinergic enzymes. Additionally, the author should extend the review on Ru complexes tested by other research groups and compare these with her findings.
It would also be beneficial if the author includes any data (if available) about other targets of Ru complexes that are important in AD.
Minor remarks:
- The author states that Ru complexes are not selective toward ChE enzymes (AChE and BuChE) in line 362, page 8. This should be clarified: are they not selective towards other enzymes involved in the pathology of AD or enzymes related to other diseases? This fact is crucial because a multi-target effect of a drug is often desired.
- The author mentions that Ru complex IC50 values are in the LOW micromolar range (line 363 and other lines). The values 1.9 and 13.14 μM are not considered low. Please delete "low."
- In line 214, the phrase “Its particular structure allows simultaneous inhibition of the active site (Tyr 337 of the anionic subsite) and PAS (aromatic π-π-stacking interactions) of AChE …” is incorrect. The active site (as well as PAS) cannot be inhibited, but the enzyme can. The author likely meant that the Ru complex binds to the active site or PAS instead of inhibiting them.
Round 2
Reviewer 3 Report
Comments and Suggestions for Authors
The author accepted all reviewer's suggestions and significantly improved her manuscript . Therefore, in this form this manuscript should be accepted for publication.